# Mechanisms of plant acclimation to multiple abiotic stresses
Zhang Jiang[1], Martijn van Zanten [1,2] ✉ & Rashmi Sasidharan [1,2] ✉

Plants frequently encounter a range of abiotic stresses and their combinations. Even though stresses rarely occur in isolation, research on plant stress resilience typically focuses on single environmental stressors. Plant responses to abiotic stress combinations are often distinct from corresponding individual stresses. Factors determining the outcomes of combined stresses are complex and multifaceted. In this review, we summarize advancements in our understanding of the mechanisms underlying plant responses to co-occurring (combined and sequential) abiotic stresses, focusing on morphological, physiological, developmental, and molecular aspects. Comprehensive understanding of plant acclimation, including the signaling and response mechanisms to combined and individual stresses, can contribute to the development of strategies for enhancing plant resilience in dynamic environments.

Climate change-related increases in global temperatures and increased incidences of weather extremes pose serious challenges for global food security[1,2]. According to a recent report published by the Intergovernmental Panel on Climate Change, climate change-associated abiotic stresses such as heat waves, droughts, floods, and storms increasingly cause massive crop losses[3]. To enhance crop resilience to erratic weather patterns, a comprehensive understanding of plant stress responses and acclimation mechanisms is essential. Current knowledge in this area is predominantly based on experimental studies using plants exposed to single stresses. However, in the field, plants rarely encounter abiotic stresses in isolation. For example, heat and drought often co-occur, and drought and flooding episodes frequently happen sequentially[4–8]. Co-occurring abiotic stresses often cause distinct effects on plants compared to individual stresses[4,9–13]. A recent meta-analysis assessing >120 published cases studying crop responses to combined heat and drought stress revealed that the combined stress caused on average, twice the decrease in yield (relative to control) compared to exposure to heat stress alone[14].

For plants grown in natural or agronomic field conditions, encountered (combined) abiotic stresses often occur at a gradual or sublethal severity, and hence are considerably mild relative to those reported in experimental laboratory studies[6,15]. Compared to severe lethal stresses and stresses at a moderate severity, mild sublethal stresses typically cause less damage to plant growth but can evoke distinct acclimation responses enabling the plant to optimize performance under the non-optimal conditions imposed by the stressor[16–18] (Fig. 1a, b).

The last decade has seen substantial progress in our understanding of plant multi-stress resilience. These studies have underscored the importance of replicating stress combinations that occur in the field and have revealed how acclimation responses are governed by highly coordinated and complex molecular networks, especially when multiple stressors coexist[19–21]. In this review, we first provide a brief summary of the current understanding of plant responses to single abiotic stresses focusing on temperature and precipitation extremes (flooding and drought), followed by a comprehensive overview of recent findings that contribute to a mechanistic understanding of how plants acclimate to combinatorial stresses. Finally, we discuss the challenges ahead and propose future perspectives for research on multi-stress responses in plants.

## Plant responses to (single) abiotic stresses

Studies on plant resilience to various environmental signals have provided indispensable insights into the molecular machinery underlying functional response strategies to diverse isolated environmental stresses[20,22–25].

**Temperature extremes.** Climate change-associated increases in average temperatures are a major threat to crop growth and yield[26,27]. Warmer temperatures in turn increase the incidence of precipitation extremes, leading to droughts and flooding events[3,28–30].

Heat stress adversely affects a variety of plant physiological processes including photosynthesis, cell membrane thermostability, and osmotic regulation[31–33]. The expression of *heat shock transcription factors (HSFs)* followed by the accumulation of chaperone heat shock proteins (HSPs) is rapidly induced upon heat perception to safeguard cells (Fig. 2). This facilitates the resumption of normal cellular and physiological activities while alleviating cell damage[34]. However, temperature increases in natural or agricultural settings are sometimes gradual and mild, involving only a few degrees of elevation within the ambient temperature range[35]. Even such mild

[1]Plant Stress Resilience, Institute of Environmental Biology, Utrecht University, Padualaan 8, Utrecht, The Netherlands. [2]These authors jointly supervised this work: Martijn van Zanten, Rashmi Sasidharan. ✉e-mail: m.vanzanten@uu.nl; r.sasidharan@uu.nl

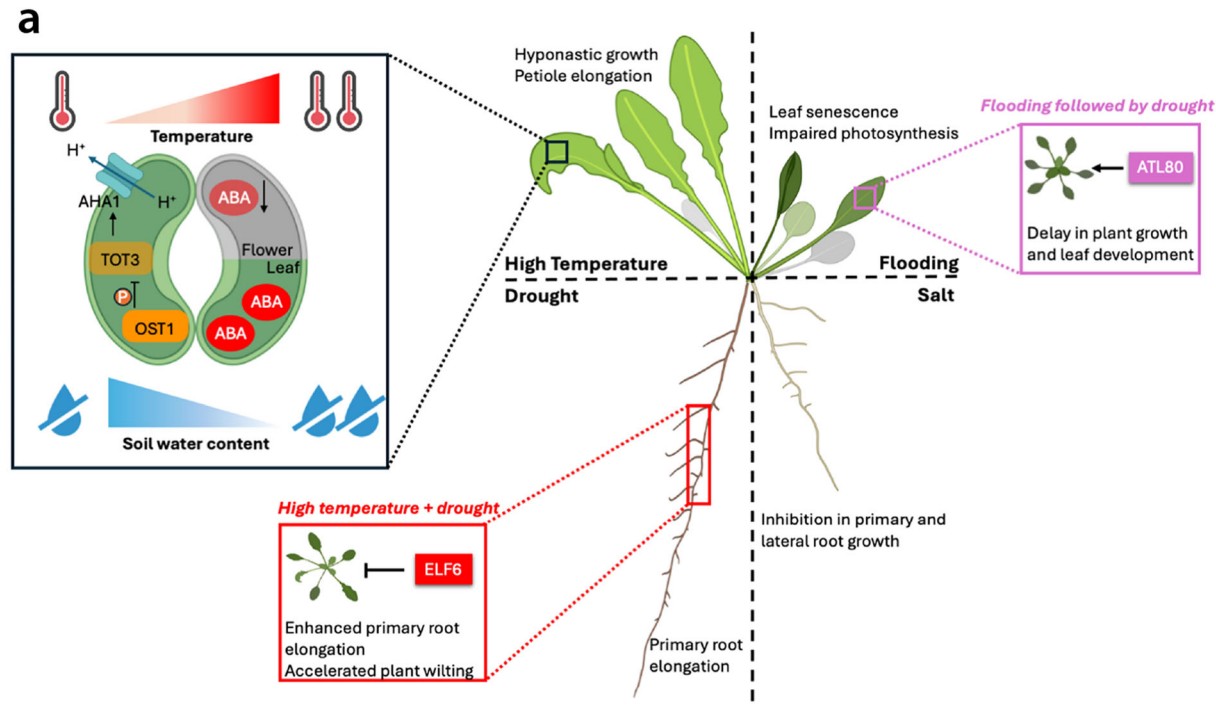

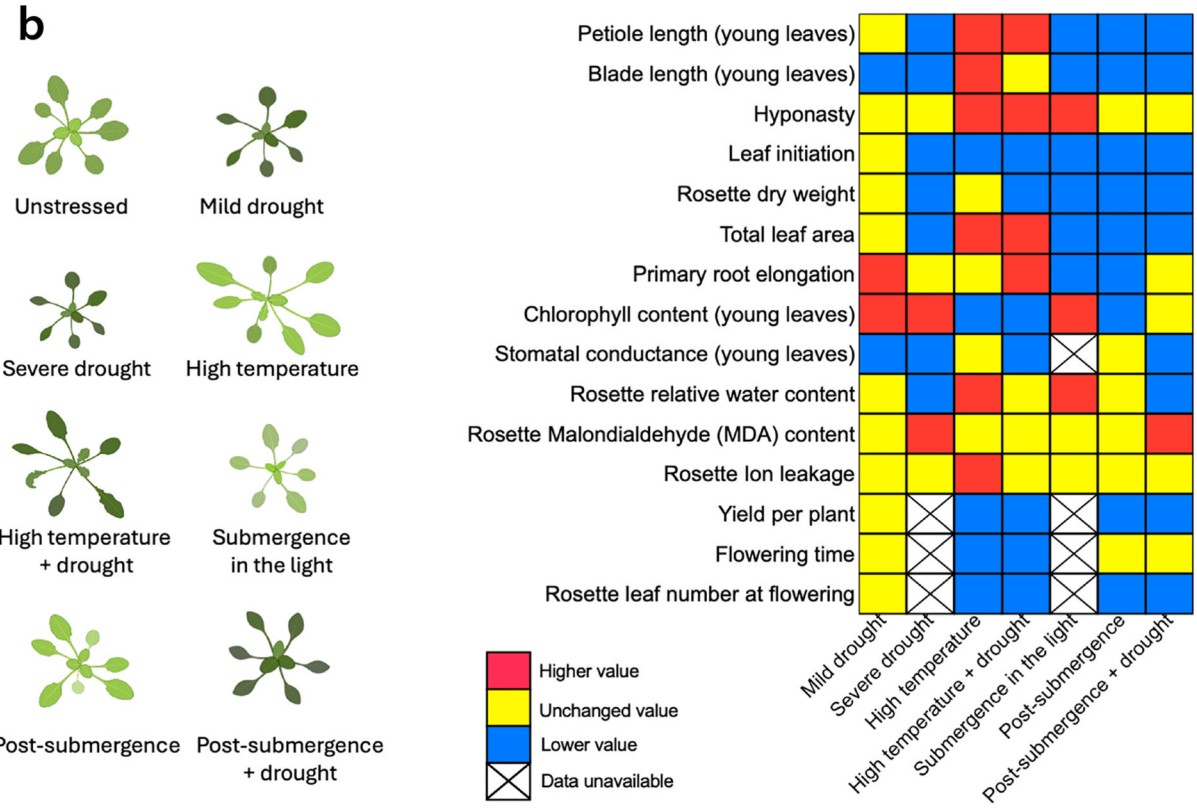

changes, when imposed on plants that are susceptible (i.e. Arabidopsis), can result in a suite of morphological alterations including hypocotyl, petiole and root elongation and hyponasty (increase in leaf angle) (Fig. 1a, b). These traits can enhance plant cooling capacity during growth in warm temperatures[36–38]. The suite of induced traits is termed 'thermomorphogenesis'. Thermomorphogenesis is considered a whole-plant acclimation strategy (or a 'trait syndrome') and is typically governed by a complex signal transduction network consisting of diverse regulatory modules[39–41]. For example, the perception of high temperature in Arabidopsis is partially accomplished by the phytochrome B photoreceptor (phyB), which directs the expression of a subset of high temperature-responsive genes[42,43]. The most comprehensively characterized signal

**Fig. 1 | Plant acclimation responses to individual and combined abiotic stresses. a** In *Arabidopsis thaliana* (Arabidopsis), high temperature promotes thermomorphogenic responses (e.g., hyponastic growth and petiole elongation), while flooding leads to inhibition of photosynthesis and eventually results in leaf senescence. Drought promotes primary root elongation, whereas prolonged salt exposure restricts primary and lateral root growth. Under combinatorial abiotic stresses, such as combined high temperature and drought, or flooding followed by drought, unique combinations of phenotypic traits emerge, which are likely orchestrated by the observed unique transcriptome signature under combined and sequential stress (as compared to the single stresses)[4]. Key regulators including EARLY FLOWERING 6 (ELF6) (red inset; for high temperature combined with drought) and ARABIDOPSIS TÓXICOS EN LEVADURA 80 (ATL80) (pink inset; for flooding followed by drought) modulate the growth, development and survival (wilting)[4,6] to combined stress. The morpho-physiological trait response values upon combined stresses are affected by the severity of each stressor and the tissue under study. For example, sublethal high temperature (27 °C), in combination with progressive drought, leads to stomatal closure through a "gas-and-brake" regulatory mechanism in Arabidopsis (black inset)[107]. High temperature activates the kinase TARGET OF TEMPERATURE 3 (TOT3), promoting stomatal opening through the H⁺-ATPase mediator ARABIDOPSIS H⁺-ATPase 1 (AHA1), while OPEN STOMATA 1 (OST1) phosphorylates TOT3 under drought to inhibit stomatal opening[107]. In soybean, heat (38 °C during the day, 28 °C at night), combined with severe drought, triggers differential stomatal regulation in different plant tissues with leaf stomatal closure controlled by Abscisic acid (ABA) accumulation, but flowers maintaining open stomata by suppressing ABA levels to protect reproductive processes[110]. These findings suggest that fundamental insight into the mechanisms regulating multi-stress acclimation obtained using the Arabidopsis model may translate to crops, though species-specific differences exist. **b** Artist impression (left; note the differences in morphology and chlorophyll content) and heatmap (right) depicting Arabidopsis acclimation responses to combined high temperature and drought, flooding followed by drought, and the corresponding individual stresses at the morphological, physiological, and developmental levels. Color values in the heatmap represent qualitative classifications of changes in trait responses under the indicated combined stresses, relative to non-stressed control conditions (red: overall higher value; yellow: unchanged value; blue: overall lower value; white crossed-out boxes: data unavailable), derived from published experiments[4,6]. Created in BioRender. Jiang, Z[4]. https://BioRender.com/l57n315.

mediator involved in thermomorphogenesis is PHYTOCHROME INTERACTING FACTOR 4 (PIF4), which acts as a master transcription factor (TF) hub regulating downstream responses[41,44,45]. Under warm temperatures, PIF4 transcriptionally activates, among other genes, the rate-limiting auxin biosynthetic gene *YUCCA8* (*YUC8*)[46], to eventually promote thermomorphogenic growth. Moreover, both PIF4 and auxin functionally depend on Brassinosteroids (BRs) as the BR-activated TF BRASSINAZOLE RESISTANT 1 (BZR1) participates in the regulation of PIF4 and growth-promoting genes during thermomorphogenic responses[47–49].

Next to PIF4, PIF7 has also been implicated as a crucial regulator of thermomorphogenic responses and is considered a bona fide thermoreceptor[50–52]. The mutual dependency of PIF7 and PIF4 possibly involves the formation of heterodimers[51]. However, in response to simultaneous warm temperature and shade, PIF7 seemingly plays a more dominant role compared to PIF4[53].

Cold stress includes chilling (>0 °C) and freezing (<0 °C) temperatures[54]. Like heat stress, exposure of plants to cold causes damage at the cellular level[54–56]. This can result in, for example, excessive production of reactive oxygen species (ROS) and lipid peroxidation and hence growth inhibition[57–60]. In Arabidopsis, cold stress can be perceived by Ca²⁺ receptors in the plasma membrane, resulting in a Ca²⁺ influx to activate downstream signaling pathways[55,61]. In response to cold, INDUCER OF CBF (C-repeat-binding factor) EXPRESSION 1 (ICE1) is phosphorylated, stimulating the expression of *C-REPEAT BINDING FACTOR* (*CBF*) genes[57,62,63]. CBFs in turn, can bind to the promoters of *COLD REGULATED* (*COR*) genes such as *COR15a* and *COR78* and activate their expression. Together, this has been termed the ICE1-CBF-COR regulon-dependent cold-stress response[64–66]. However, *COR* genes can also be regulated by the phytohormone Abscisic acid (ABA), independent of *CBFs*[57,67].

**Salt and drought**. Both salt and drought impose plant turgor loss at high concentrations[68]. The early responses to salt are closely related and mechanically overlap with drought responses, as they both elicit osmotic stress[69] (Fig. 1a). However, prolonged exposure to salt leads to toxicity and nutrient imbalance in addition to water limitation[70,71]. Stomatal closure is a typical physiological response imposed by both salinity and drought to prevent transport-mediated water loss, despite photosynthesis being disrupted due to impaired gas exchange[72,73]. The regulation of stomatal closure during salinity or drought is primarily controlled by ABA through a series of signaling components in guard cells such as ROS, reactive carbonyl species (RCS), nitric oxide (NO) and Ca²⁺[74–76]. ABA has also been demonstrated to have a prominent role in regulating root growth and architecture under salt and drought conditions[77]. When Arabidopsis plants encounter moderate to high salt concentrations (75-150 mM NaCl), the elevation of endogenous ABA levels results in a quiescent period in post-emergent lateral roots, forming Casparian strips (a ring-like, specialized cell-wall modification) that function as a barrier to the diffusion of sodium (Na⁺) ions through the endodermis[77–80]. ABA-mediated root responses during drought involve primary root elongation (Fig. 1a). Upon moderate drought, ABA promotes auxin transport in the root tip of Arabidopsis and rice (*Oryza sativa*), enhancing the release of protons by activating H⁺-ATPase proton pumps to maintain primary root elongation. This permits subsoil foraging for water and nutrients, that ultimately enables restoration of hydraulic conductivity[75,81,82].

**Flooding**. In contrast to drought, flooding (root waterlogging or partial or whole plant submergence) creates an excess water supply. The aqueous environment disrupts normal gas exchange and may reduce light availability (when shoots are submerged in turbid floodwaters). The resulting impairment of aerobic respiration and photosynthesis leads to a carbon and energy crisis and ultimately cell death[83,84] (Fig. 1a). When plants are flooded, the limitation in gas diffusion also causes rapid accumulation of the volatile phytohormone ethylene. Ethylene is a key player mediating a series of flood-adaptive morphological and physiological changes in both shoot and roots[85,86]. Typical underwater responses triggered by ethylene accumulation include accelerated petiole (*Rumex palustris*) or internode (*Oryza sativa*) elongation upon complete submergence[87–90], or the development of aerenchyma during waterlogging. Both these traits facilitate enhanced internal aeration, permitting gas exchange from aerial non-flooded parts to the hypoxic regions of the plant[91–94]. In Arabidopsis, ethylene accumulation due to flooding leads to stabilization of the group VII ETHYLENE RESPONSE FACTORs (ERFVIIs) TFs, through NO depletion, which consequently results in hypoxia acclimation[95,96]. In deepwater rice, ethylene accumulation induces the expression of *SNORKEL1* and *SNORKEL2*, promoting internode elongation via gibberellin phytohormones[97,98]. In contrast, the ethylene-inducible ERF VII TF, *SUBMERGENCE-1A* (*SUB1A*), restricts growth and energy utilization and confers tolerance to multiple stresses including drought, submergence and dehydration experienced upon de-submergence in rice[99]. In addition to tissue dehydration, plants recovering from submergence in darkness also encounter challenges such as reoxygenation stress, reillumination stress and senescence[100–102].

**Effects of combined abiotic stresses on plant functioning**
Despite a typical generic reduction in growth and yield when faced with combinatorial stresses[14,103,104], plants are not passive and have evolved a series of adaptive responses at the morpho-physiological level to counteract unfavorable combined stress conditions[10,13,24]. The exact nature of the responses to a given combinatorial stress often differs from those elicited by the corresponding individual stressors, as plants perceive the stress

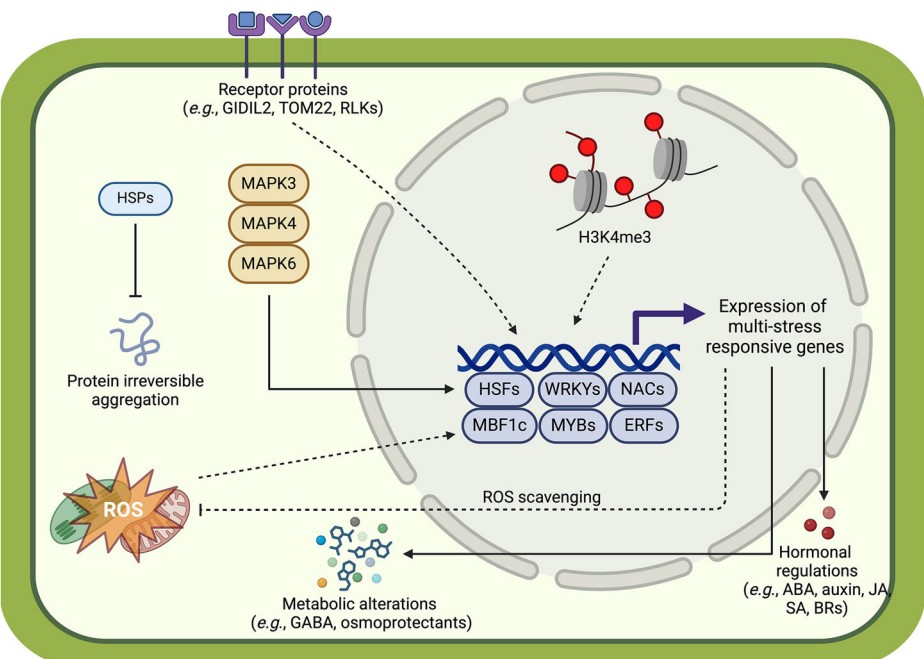

**Fig. 2 | Molecular mechanisms potentially involved in combinatorial abiotic stress perception and signaling.** Schematic overview of confirmed (solid lines) and putative (dotted lines) sensing and signaling pathways proposed to mediate acclimation to combinatorial abiotic stresses, based on knowledge of single stress signaling and perception pathways. At the transcriptional level, key regulators, including TFs from the Multiprotein Bridging Factor 1c (MBF1c), Heat Shock Factor (HSF), MYELOBLASTOSIS (MYB) (all implicated in stress combination: heat and drought, see main text), WRKY (combined high light and heat stress, multifactorial stress), NAM, ATAF1/2, and CUC2 (NAC) (all associated with diverse multi-stress pathways) and ETHYLENE RESPONSIVE FACTOR (ERF) families (drought, submergence and dehydration stress, and possibly combined drought and heat, salinity and heat, and high light and heat), mediate the expression of multi-stress responsive genes, leading to e.g. hormonal and metabolic alterations. At the translational and post-translational levels, HSPs mitigate protein aggregation by chaperoning stress-denatured proteins and facilitating the transport of unfolded or misfolded proteins out of the endoplasmic reticulum (combinations of heat stress, high irradiance, and drought). MITOGEN-ACTIVATED PROTEIN KINASES (MAPK) signaling, through e.g. MAPK3, MAPK4, and MAPK6, (heat and salinity stress) activate downstream targets to counteract combined heat and salt stress. Receptor proteins such as GIDIL2, TOM22 (heat and drought), and Receptor-like kinases (RLKs; various types of stresses via ROS and ABA signaling), are proposed regulators of combinatorial-stress acclimation. At the epigenetic level, tri-methylation of histone H3 at lysine 4 (H3K4me3; high temperature combined with salinity and/or drought), which is known to control plant thermomorphogenic responses, likely contributes to combinatorial-stress responses. Hormones such as abscisic acid (ABA), auxin, jasmonic acid (JA), salicylic acid (SA), and Brassinos-teroids (BRs), are essential in regulating plant resilience to combined stresses. Reactive oxygen species (ROS), as a driver of plant systemic signaling during combinatorial stress responses, are at the core of multi-stress signal integration, tuning response severity and likely mediating retrograde signaling between plant organelles (e.g., chloroplasts and mitochondria) and the nucleus. Created in BioR-ender. Jiang, Z. (2025) https://BioRender.com/o02r418.

combination as a new state of stress[4,105,106]. Therefore, effects of combinatorial stress cannot be deduced by simply summing up the effects of the corresponding single stresses.

**Stomatal responses.** A well-documented example of specific physiological output determined by interaction between multiple stresses is the stomatal response of Arabidopsis to combined heat and drought stress[7] (Fig. 1a). When confronted with heat stress, Arabidopsis plants open their stomata to enable leaf cooling through transpiration, while under drought they reduce stomatal conductance to prevent water loss. Upon the simultaneous application of both heat and drought, stomatal conductance typically remains at a low level[7]. Fine-tuning the stomatal responses to conflicting heat and drought signals is controlled by a gas-and-brake-like mechanism involving complex molecular regulations[107]. Similar results of the leaf stomatal response under combined heat and drought stress conditions were observed in other plant species such as tobacco (*Nicotiana tabacum*)[108], citrus plants (*Citrus medica*)[109], various broadleaf evergreen species[30] and soybean (*Glycine max*)[110], suggesting that stomatal behavior under combined heat and drought is conserved. However, recent studies[11,111,112] have unveiled 'differential transpiration' regulation in soybean and tomato flowers subjected to a heat and drought combination (Fig. 1a). In these intriguing case examples, plants prioritized transpiration through flowers over leaf transpiration, to ensure a lower innate temperature of the floral structures and maintain proper functioning of reproductive processes. Consequently, when heat and drought co-existed, leaf stomata remained closed, while flower stomata were open. The differences in stomatal regulation within the same individual highlight the complexity of (tissue-dependent) responses to combinatorial stresses. It must be noted that the age or developmental stage at which plants are exposed to combinatorial stresses also determines the outcome on plant growth and morpho-physiological responses[4].

**Photosynthesis.** Stomatal aperture is an important determinant of $CO_2$ uptake capacity. In addition, stress exposure can have drastic effects on chloroplast structures, but also on resident proteins processing $CO_2$, such as ribulose bisphosphate carboxylase oxygenase (Rubisco) and Filamentous temperature sensitive H (FtsHs)[113–116]. Stomatal aperture, together with the stabilization of Rubisco and the functional integrity of Photosystem II (PSII) affect photosynthetic activity during combinatorial stresses[117–119]. Combined heat and drought stress, for example, exacerbated impairment of photosynthesis in both C3 and C4 plants, compared to either heat or drought stress experienced in isolation[120,121]. In cotton (*Gossypium arboreum*) cultivars, combined heat and drought caused a decrease in net photosynthetic rate and hindered leaf development[122,123]. This inhibition of photosynthesis, reflected by a low Rubisco activity, was more pronounced in a drought-sensitive cultivar compared to a drought-tolerant one[122]. However, a significant decrease in

photosynthetic activity was observed in heat-tolerant tomato (*Solanum lycopersicum*) cultivars, but not in a heat-sensitive cultivar, during simultaneous exposure to combined heat and drought stress[124]. This suggests that photosynthetic responses to combined stresses does not necessarily align with tolerance to individual stresses. These studies imply that the maintenance of photosynthetic activity is important for acclimation to combinatorial stresses[123] and is likely genotype-dependent.

### ROS production and signaling

Chloroplasts play a central role in sensing environmental fluctuations[125]. The chloroplast is a major source of ROS production during native photosynthesis and especially under stress conditions[126,127]. It has been proposed that combined ROS production from the chloroplast and other cellular compartments (e.g., mitochondria, apoplast, peroxisome, nuclei) participates in quantitative stress sensing (Fig. 2). In this model, total ROS levels determine response strength to different stressors, while stress-specific signaling components determine which precise pathways are activated[106,128]. Moreover, cellular ROS dynamics could constitute specific 'ROS signatures' that can vary between single and combined stresses[9,129]. For example, in poplar (*Populus yunnanensis*) plantlets, different ROS levels caused by individually applied heat, drought and the combination resulted in varying levels of antioxidant enzyme production[130].

Disrupted photosynthetic capacity, accompanied by limited $CO_2$ availability, can induce ROS over-production and eventually cause damage to lipid membranes and cellular organelles[117,129,131,132]. However, ROS over-production can be ameliorated by the accumulation and activation of ROS detoxification proteins such as superoxide dismutase (SOD), ascorbate peroxidase (APX), peroxiredoxin and glutathione peroxidase (GPX), or antioxidants such as glutathione (GSH) and ascorbic acid[129,133,134]. These ROS-scavenging components display a unique pattern (in terms of types of enzymes and levels of antioxidant accumulation) under combined stresses compared to the relative individual stresses[9]. High antioxidant capacity under stress conditions is considered beneficial for stress tolerance, as it limits damage[135–137], and is typically genotype-dependent[138]. Differences in antioxidant capacity, for instance, explain the differences in drought and high temperature stress tolerance between two citrus genotypes, Carrizo citrange (*Citrus sinensis* × *Poncirus trifoliata*) and Cleopatra mandarin (*Citrus reshni*). In this study, Carrizo plants coordinated the antioxidants involved in ROS detoxification more efficiently and displayed a better performance and yield under combined stress than Cleopatra plants[132].

In addition to quantitively determining stress response levels, ROS also have a signaling role, especially during simultaneously occurring stresses[123,131,139] (Fig. 2). Zandalinas et al. showed that Arabidopsis plants with impaired ROS signaling (*rbohD* mutant) or scavenging (*apx1* mutant) exhibited poor survival rates under multifactorial stress combinations (up to a combination of six stressors at the same time), compared to wild type plants[140]. Furthermore, ROS waves have been identified as essential drivers of plant systemic signaling pathways in response to combinatorial stresses such as combined high light and heat stress or sequentially applied waterlogging followed by complete submergence[141,142]. Another indispensable role for ROS in plant stress acclimation is the modulation of signal communications between chloroplast and nucleus (retrograde signaling)[143,144]. For instance, chloroplast-localized ETHYLENE-DEPENDENT GRAVITROPISM-DEFICIENT AND YELLOW-GREEN 3 (EGY3) interacts with Cu/Zn-SOD2 (CSD2) to promote $H_2O_2$-mediated retrograde signaling, enhancing the salt tolerance of Arabidopsis plants[145]. However, ROS-mediated chloroplast-to-nuclear retrograde signaling in the context of combinatorial-stress tolerance has been poorly studied so far.

### Concepts of stress interactions

In wheat (*Triticum aestivum*), episodes of prolonged drought combined with heat waves exacerbate biomass reduction and loss of grain yield compared to individually applied drought or heat[146–148]. This inhibition in growth and yield under combined stress is attributed to the negative interactions between heat and drought, with the effects becoming additive when combined[20]. Next to additive (negative) effects elicited by stress combinations[4,149,150], the co-occurrence of two different environmental stressors can sometimes lead to antagonistic effects[20], which may stimulate resilience to one or both stressors. For example, drought-induced reduction in stomatal conductance can enhance the tolerance to ozone ($O_3$) stress, when the two stresses co-occur, as the closed stomata prevent $O_3$ from entering the plants[151,152]. In 2006, Mittler and colleagues introduced the concept of 'The Stress Matrix' to describe the interactions of (two) co-existing stressors in stress combinations that have significant impacts on agricultural production[153]. However, this matrix may oversimplify the complexity of combined stress scenarios[154]. In recent years, the stress matrix has therefore been adapted and expanded to include more stress interactions[13,155,156] and different physiological factors[11,13]. Suzuki and coworkers[20] further refined the stress matrix by considering, for example, the dual interactions (both positive and negative) of combined heat and salinity stress on plant growth. They demonstrated that combined heat and salinity promote the accumulation of glycine betaine and trehalose in tomato plants. This helps in maintaining a high $K^+$ concentration (thus a lower ratio of $Na^+$ and $K^+$) and improves cell water status and photosynthesis compared to salinity alone[157]. However, the same combined stress scenario evoked enhanced negative effects on tissue development in wheat seedlings[158] or photosynthetic growth in Arabidopsis[159].

In general, the physiological (and molecular) response to combinatorial stress is predominantly determined by the (relative) most severe stressor[105]. Accordingly, the magnitude, order and duration of the two stressors are crucial in determining the morpho-physiological outcomes of combinatorial stresses[10,153,155]. Stress magnitude, or 'dose', refers to the relative or absolute intensity of individual stressors, such as the absolute temperature during heat or cold stress[37,160], the soil water content during drought stress[161], or light availability during submergence[17,84,162]. Additionally, the number of co-existing stressors during a combinatorial stress scenario is also an important factor in determining the outcome for the plant. Recent studies presented the (above-mentioned) new concept of 'multifactorial stress combination' to describe how the combination of many co-occurring environmental stresses (up to six) affects plant growth, survival, physiological and molecular responses[140,163,164]. These studies suggested that, while individually applied abiotic cues sometimes have minimal -or no - effect on plant growth and survival, the accumulated impact of these cues can become cumulatively stressful and detrimental. This highlights the synergistic interactions among individual stressors when they occur simultaneously.

Building upon these findings and some other pioneering studies discussing how global change factors impact ecosystem processes[165,166], a 'multifactorial stress principle' was proposed to depict the synergistic effects of stapled/accumulating stressors/cues and how they affect individual plants and ecosystems[11]. With an increase in the number and complexity of stressors (simultaneously) affecting a plant or an ecosystem, plant functioning or ecosystem processes will drastically decline, even if the level of each of the individual stressors involved in the multifactorial stress combination is low enough to not significantly affect plant growth and survival if applied in isolation. Overall, these studies emphasized the importance of considering the relevance and impact/magnitude/dose effect of subtle (or sublethal) stresses when studying plant acclimation to combinatorial stresses.

While the responses to combinatorial stresses are often largely determined by the most severe stressor, the order of the events also matters. When plants are confronted with a sequential stress combination, the first stress exposure, even if mild, may induce priming or memory effects, altering the responses to future challenges[12,167], referred to as cross acclimation. Therefore, the order in which the two stresses are encountered can be crucial in determining the effect size of the plant response[140,149]. For instance, poplar plants that were pre-exposed to drought exhibited a reduction in stomatal conductance, which alleviated the harsh effect of a subsequent $O_3$ stress, as indicated above. Conversely however, when $O_3$ stress was applied prior to drought, the slow stomatal responses induced by

$O_3$ accelerated plant water loss during a subsequent drought exposure[154,168]. The molecular mechanisms of cross-acclimation due to priming or memory effects remain poorly understood but is a promising direction for future research aimed at enhancing the resilience of economically important crops.

## Molecular mechanisms underlying acclimation to combinatorial stresses

Morpho-physiological responses to sequential and combined stresses often involve changes at the transcriptional, translational, and metabolic levels and are coordinated by complex integrated signal transduction networks[12,155]. Combinatorial stresses can elicit unique molecular signatures that are different from those induced by either of the corresponding individual stresses, as has been shown by many ~omics studies[4,7,10,11,116,123,138,153]. However, in line with the notion that plant responses to a given stress combination are predominantly determined by the more severe (dominant or first-experienced) stressor, the transcriptome response to combined stress often resembles that of the more severe individual stress[105]. The similarities at the molecular level are then reflected by a substantial proportion of shared transcripts. But also, many unique genes can be regulated that are not affected by application of either of the single stresses. For example, a profound transcriptome reconfiguration was detected during Arabidopsis exposure to sublethal co-occurring high ambient temperature and drought stress relative to the corresponding individual stresses, with high temperature having the largest effect on the response[4]. Although there was considerable overlap between high temperature-regulated genes and those affected by combined high temperature and drought, most of the genes responsive to the combined treatment were not affected by either drought or temperature alone. Conversely, not all genes affected by drought or high temperature in isolation were affected when the treatments were combined[4].

Plants can also harmonize the conflicting signals evoked by coinciding stresses. A recent study presented a 'gas-and-brake' mechanism controlling stomatal aperture during co-occurring high temperature and drought in Arabidopsis[107] (Fig. 1a). Under high temperature conditions, the protein kinase TARGET OF TEMPERATURE 3 (TOT3) activates ARABIDOPSIS $H^+$-ATPase 1 (AHA1) to promote stomatal opening. However, when drought accompanies high temperature, TOT3 is deactivated through OPEN STOMATA 1 (OST1)-mediated phosphorylation, resulting in stomatal closure[107].

Systems biology approaches involving the integration of more than one ~omics dataset (commonly referred to as 'Multi-Omics Approach'), have emerged as a tool to comprehensively decipher molecular acclimation strategies of plants under combinatorial stresses[13,138,155,169–171]. Anwar et al.[155] summarized previously identified response characteristics of Arabidopsis and maize plants under heat, drought and their combination at the transcriptomic, proteomic, and metabolic levels[7,116,155]. This study revealed a significant number of differentially regulated transcripts, proteins, and metabolites under the combined stress that were not apparently regulated if only the corresponding single stresses were applied. This again underlines the notion that combined stress exerts unique and significant reconfigurations at different molecular levels. Recently, a publicly available platform, the Stress Combinations and their Interactions in Plants Database (SCIPDb), was developed to facilitate a comprehensive understanding of plant responses to combinatorial stress, including both simultaneous and sequential stress occurrences[13]. SCIPDb integrates data from over 900 studies, including phenotypic (morphological, physiological, and biochemical) and molecular (transcriptomic and metabolomic) aspects of plant stress responses[13]. This valuable resource has an interactive platform allowing users to search for specific stress combinations, visualize relevant datasets and access analytical tools for preliminary data interpretation.

Such resources and studies, will be critical for gaining useful insights into the typical generic processes and master regulators safeguarding plants against negative effects of combinatorial stresses[4,21,105,106,172].

**Transcriptional regulation**. Transcription factors (TFs) are essential for controlling growth and developmental processes that shape acclimation to environmental stimuli and mediate responses to combinatorial stresses[13,173–175] (Fig. 2). An early study by Suzuki et al. highlighted a transcriptional coactivator, Multiprotein Bridging Factor 1c (MBF1c), and its function in conferring tolerance to osmotic stress, heat stress and their combination[176]. In this context, MBF1c perturbs or partially activates the ethylene-response signal transduction pathway[176]. Functional characterization of MBF1c revealed that its accumulation in Arabidopsis plants under combined water deficit and heat stress is ABA-dependent[177].

Investigations into plant TFs and their functions in combinatorial-stress acclimation often take several members of the TF family into account (Fig. 2). For example, the NAC (NAM, ATAF1/2, and CUC2) TF family, one of the largest plant TF families in existence, has been implicated in regulating multi-stress tolerance in various plant species[178–180]. By examining the Arabidopsis transcriptome under combined heat and drought stress and the corresponding individual stresses[7], a considerable number of transcripts encoding Heat Shock Factors (HSFs) were enriched during the stress combination and were differentially regulated compared to the corresponding individual stresses. The differences mainly involved the degree of expression of *HsfC1* and the presence of *HsfA6a, HsfA2*, and *HsfA3* transcripts[7]. A meta-analysis[21] identified 340 transcripts that were commonly upregulated during Arabidopsis subjection to combined drought and heat[7], salinity and heat[159], and high light and heat[113]. Among these transcripts, TFs belonging to the HSF, MYELOBLASTOSIS (MYB) and ETHYLENE RESPONSIVE FACTOR (ERF) families were significantly overrepresented. Moreover, the distinct expression patterns of these TFs under combined stress – as compared to individual stresses – suggest that plant transcriptomic responses to each stress combination maybe regulated by unique, dedicated TFs. This can be by means of additive, subtractive or combinatorial effects of expression (patterns) of different groups of TFs, generating a distinct overall TF signature that is unique to the stress combination and severity (Zandalinas et al.[21]). Although some studies demonstrated that WRKY TFs play a role in acclimation to combined abiotic and biotic stresses[181], none were present in the 340 transcripts identified by (Zandalinas et al.[21]). However, more recent studies on plant responses to multifactorial stress combinations have drawn attention to the role of WRKYs in conferring plant responses to four- to six- factor stress combinations[140,182]. In addition, WRKY48 was recently identified as a negative regulator of plant acclimation to combined high light and heat stress in *Arabidopsis thaliana*[183]. Finally, a recent study implicated the TFs EARLY FLOWERING 6 (ELF6) and ARABIDOPSIS TÓXICOS EN LEVADURA 80 (ATL80) in mediating acclimation to combined high temperature and drought and flooding followed by drought in Arabidopsis, respectively[4] (Fig. 1a).

Given the significant role TFs play in stress acclimation, it is likely that additional TFs and their family members, are involved in regulating acclimation to combinatorial stresses, and much remains to be discovered.

In addition to the study of TF families, recent work by Azodi et al. proposed the use of *cis*-regulatory codes[184] to improve the understanding of transcriptional regulation under combinatorial stress[185]. By integrating information on putative/known combined-stress *cis*-regulatory elements and ~omics data (including sequence conservation, chromatin accessibility, and histone modification profiles), relevant *cis*-regulatory promoter elements mediating tolerance to combined heat and drought stress were predicted. While most of the *cis*-regulatory elements found in the model are similar to known TF binding motifs involved in heat and/or drought stress responses, some point to TFs with no established association to either stress condition[185]. Likewise, another study[186] on transcriptional and metabolic responses to drought, heat, salinity, and their combinations demonstrated that plant exposure to combinatorial stress conditions triggers the transcription of several genes with as yet uncharacterized functions. Overall, these findings highlight the complexity of transcriptional regulation in plants under combinatorial stresses and indicate that current knowledge on this important subject still needs to be expanded.

**Post-transcriptional regulation**. Transcriptomics studies have provided valuable insights into molecular regulation of multi-stress acclimation. However, transcriptional and translational responses do not always correlate[187,188], making it crucial to expand investigations to other regulatory layers. Indeed, post-transcriptional regulation, particularly at the microRNA (miRNA) level, has emerged as a key factor in the modulation of stress signaling pathways[12,189] (Fig. 2). In recent years, a growing body of evidence on the role of plant miRNAs as (a)biotic stress regulators, has added new conceptual insights into the molecular understanding of plant stress resilience[190,191]. However, research investigating the regulatory roles of miRNAs during combinatorial stress is still relatively scarce[189,192]. Nonetheless, some researchers have taken the initiative to explore this field. For example, by taking a deep-sequencing approach unique miRNAs and their targets were found to be uniquely associated with combinatorial stress conditions in different plant species such as tomato[193], soybean[194], and melon[192]. miRNAs were also shown to be closely associated with the regulation of specific biological processes under combinatorial stresses. For instance, an Arabidopsis loss-of-function miRNA mutant *ath-miR164c* exhibited proline accumulation to counteract harsh effects caused by combined drought stress and bacterial infection[195] (Fig. 2). This was due *ATH-miR164C*-mediated negative regulation of the expression of *1-PYRROLINE-5-CARBOXYLATE SYNTHASE 1* (*AtP5CS1*), a gene that controls proline metabolism, at the post-transcriptional level. In addition, Liu et al. recently constructed a comprehensive regulatory network that illustrated the molecular responses to combined heat and drought in durum wheat (*Triticum turgidum durum*), by integrating multiple ~omics analyses, including assessment of the small RNAome (sRNAome), mRNA transcriptome, and degradome[196]. This study provides fundamental insight into transcriptional and post-transcriptional regulation of combinatorial stress at the whole-genome level.

**Translational and post-translational regulation**. Abiotic stresses can have a significant impact on the plant proteome, especially when multiple stresses coincide[197]. This is evidenced by numerous differentially regulated proteins detected uniquely under combinatorial stresses compared to the relative individual stresses[7,114,130,198]. As noted before, Heat Shock Proteins (HSPs) are among the most prominent proteins regulating plant tolerance to various combinatorial stresses[7,21,116,130,186]. A recent study[199] investigating proteome and transcriptome signatures of before-mentioned citrus genotypes, Carrizo citrange and Cleopatra mandarin, under the triple combination of heat, high irradiance, and drought, revealed the importance of maintaining HSPs, typically small HSPs and HSP70s, for combined stress tolerance. This is because HSPs chaperone stress-denatured proteins to prevent their irreversible aggregation and translocate unfolded or misfolded proteins out of the endoplasmic reticulum (ER)[199]. Zhao et al. found that, in addition to HSPs, LATE EMBRYOGENESIS ABUNDANT Proteins (LEAs) were also highly abundant when maize (*Zea mays*) plants were subjected to combined heat and drought stress[116]. This study also investigated the changes in receptor proteins, protein kinases, and phosphatases during combined stress conditions. When maize plants were exposed to combined heat and drought, the expression of three membrane receptor proteins was significantly regulated. This included two downregulated receptors: brassinosteroid LRR receptor kinase and gibberellin receptor GIDIL2, and the upregulated receptor: mitochondrial import receptor subunit TOM22 (Fig. 2). This points to the involvement of phytohormonal perception and regulation in response to combined stress.

Receptor-like kinases (RLKs) constitute a family of membrane receptors responsible for perceiving different environmental stimuli and balancing plant growth and stress responses[200,201]. For instance, the pathogenesis-related 5 (PR5) RLK 2 (PR5K2) has been shown to modulate plant responses to drought by phosphorylating protein Phosphatase 2Cs (PP2Cs) in Arabidopsis[202]. PP2Cs are important protein phosphatases in the ABA signaling pathway[203] and mediate ROS[204] signaling. Because both ABA and ROS signaling regulate the tolerance to various types of stresses[9,159,177], including combinatorial stresses, RLKs are promising targets for future investigations into combinatorial-stress acclimation (Fig. 2).

Other protein kinases implicated in combined stress acclimation[12,172] include mitogen-activated protein kinases (MAPKs) and calcium-dependent protein kinases (CDPKs) (Fig. 2). Noticeably, MAPK3, MPK4, and MPK6 phosphorylate the heat stress factor HSFA4A and activate the expression of the downstream targets to counteract combined heat and salinity stress in Arabidopsis[205].

**Epigenetic regulation**. Epigenetic processes involving DNA methylation and histone modifications play a crucial role in modulating the expression of stress-responsive genes by changing their chromatin status[206,207]. Extensive studies have been conducted on epigenetic and epigenomic responses to single abiotic stresses[207–209]. However, studies focusing on epigenetic regulation under stress combinations remain limited.

It is known that mildly elevated ambient temperature enables tri-methylation of histone H3 at lysine 4 (H3K4me3) and thereby promotes the expression of auxin-related genes in Arabidopsis[40]. When combined with another stress (e.g., drought, salinity), the enhanced expression of auxin-related genes may exert an additive effect on responses to the second stressor. Nevertheless, as molecular responses to combinatorial stresses are frequently distinct from those induced by the corresponding single stress[12,105,142,153,155], it is equally likely that the co-existence of two stressors may induce – and be regulated by - a unique epigenomic signature that is distinct from the one evoked by the respective single stresses.

As a result of cross-stress priming, when confronted with the second stress, the established shared signaling pathways between the two stresses facilitate the responses to the subsequent stress[210]. Cross-stress acclimation is closely associated with epigenetic regulation[211,212]. After being primed by the initial stress, plants can establish a cross-stress memory through epigenomic (including sRNA-mediated regulation, DNA methylation and chromatin changes), but also via transcriptomic, proteomic, and metabolic processes[210]. For instance, MAPK3 and MAPK6 kinases are essential for the regulation of cross-stress acclimation[213]. The epigenetically-imprinted stress memory may be inherited over generations under certain conditions (trans- or intergenerational memory)[214–216]. This could be an interesting lead towards the development of training methods that aim at enhancing crop tolerance to multiple (sequential) environmental stimuli by cross-stress priming over generations.

**Metabolic regulation**. Plant responses to combinatorial stresses often directly or indirectly involve changes in plant metabolism[123]. Metabolic profiling (metabolomics) studies have revealed additive metabolic reconfigurations in response to combinatorial stresses compared to corresponding individual stresses[7,186,217–219], and the roles of compounds in modulating diverse types of combinatorial stress acclimation have been identified using multi-omics approaches[7,186,217,219,220].

Zandalinas et al. summarized the changes in primary metabolites in Arabidopsis plants exposed to multiple individual stresses and their combinations, including changes in sugars, amino acids, tricarboxylic acid (TCA) cycle metabolites, and other molecules such as L-ascorbate and lactate[221]. When encountering environmental stresses, sugar levels in plants are drastically affected due to stress-imposed changes in photosynthesis and carbohydrate consumption[222]. However, sugars are also key players in stress perception as signaling molecules, osmoprotectants, and in ROS scavenging[223,224]. For example, Arabidopsis plants exposed to combined heat and drought stress accumulated high levels of sugars like sucrose, maltose, and glucose[7], which can function as osmoprotectants. Similarly, in maize plants exposed to combined cold and drought stress, increased raffinose levels facilitated osmotic adjustment and protection of the photosynthetic apparatus against oxidative damage[217].

Protein degradation during stress leads to the accumulation of free amino acids[225]. This also contributes to osmotic adjustment and ROS

scavenging[221,225,226]. Especially proline, a crucial amino acid, is well known for its role in maintaining proper cellular osmotic potential during stress and recovery[227]. Proline accumulated in peanut (*Arachis hypogaea*) under stress combinations that included salt as one of the component stresses (e.g., heat and salt, drought and salt, heat and salt and the osmoticum mannitol)[220]. Despite its importance in maintaining homeostasis during osmotic stresses such as salt and drought[228], proline was replaced by sucrose as a major osmoprotectant when drought coincided with heat in Arabidopsis. It is proposed that this occurs because proline might become too toxic to cells during the combined stress condition[7]. Proline content in Arabidopsis plants also increased when drought was combined with *Turnip mosaic virus* (TuMV) infection[229]. Such dynamic changes in proline accumulation point to the complexity of metabolic responses to different stress combinations.

A study by Balfagón et al. highlighted the importance of another amino acid, γ-Aminobutyric acid (GABA) in response to combined heat and high light stress in Arabidopsis, as GABA may promote autophagy during such combinatorial stresses[230]. Levels of TCA cycle metabolites decreased in plants exposed to drought in combination with salt and high light combined with heat[218,230], as these detrimental stress combinations can compromise plant respiration. In field-grown maize plants, the levels of TCA cycle metabolites negatively correlated with grain yield under combined heat and drought stress[231].

When confronted with cold combined with salt, pepper plants (*Capsicum annuum* L.) accumulated more flavonoids compared to the relative individual stresses[232]. Moreover, some studies[233–237] proposed a correlation between combinatorial stress tolerance and high levels of plant flavonoid accumulation. These findings suggest that flavonoid contributes to combinatorial stress tolerance, though further research is needed to clarify its underlying molecular mechanisms and potential for crop improvement.

**Hormonal regulation.** Phytohormone biosynthesis, degradation and signaling precisely regulate plant growth, development responses to different types of stresses[238–240]. ABA is deemed particularly important for regulating tolerance to multiple abiotic stresses, especially osmotic stresses[241,242] (Fig. 2). For instance, rice Nine-cis-epoxycarotenoid Dioxygenase 3 (OsNCED3), a gene controlling ABA biosynthesis, is responsible for conferring plant tolerance to salt, polyethylene glycol (PEG), and $H_2O_2$[243]. Overexpression of *OsNCED3* also enhanced salinity and water stress tolerance. ABA also has a role in mediating plant tolerance to combinatorial stress, particularly when an osmotic stress (imposed by drought or salinity) is one of the co-occurring stresses. For example, Arabidopsis ABA signaling or biosynthesis mutants exhibit impaired acclimation to combined heat and drought[177] and heat combined with salt stress[159], which is reflected by reduced growth and survival compared to wild type plants[159,177]. However, given the complex nature of hormonal regulation under combinatorial stresses, the alterations of applied stressors or the plant species/genotype can lead to very distinct and unique hormonal responses[221,239]. For example, in contrast to ABA being crucial for regulating heat and drought responses[177], jasmonic acid (JA) is required for Arabidopsis acclimation to combined heat and high-light stress[113]. Citrus plants subjected to combined heat and drought stress accumulated high levels of salicylic acid (SA) compared to the corresponding individual stresses and controls, while ABA levels surprisingly decreased[109]. Seemingly, this phenomenon is due to the interactions between different hormones under a specific (combinatorial) stress condition[12,239]. In a recent study, Xu et al. demonstrated a crucial role for ABA in balancing stomatal regulation under combined heat and drought stress[107]. Under heat stress, activated TOT3 phosphorylates and regulates $H^+$-ATPases, promoting stomatal opening to facilitate transpiration and cooling. However, when drought co-occurs with heat, ABA accumulation triggers *OST1* expression, which phosphorylates and inhibits TOT3, leading to stomatal closure to conserve water (Fig. 1a). Additionally, TOT3 also regulates brassinosteroid-dependent hypocotyl elongation in response to high temperature in darkness by modulating BZR1 activity[244].

Next to regulating stomatal activity, ABA also interacts with other hormonal pathways to modulate plant growth under combined stress. For example, ABA suppresses thermomorphogenic responses by counteracting auxin accumulation induced by high temperatures[245]. This crosstalk likely explains why leaf elongation is significantly repressed under combined heat and drought stress compared to heat stress alone[4,6] (Fig. 1b).

Arabidopsis mutants deficient in glutathione exhibit increased susceptibility to combined cold and osmotic stress, with a differential regulation of transcripts responsive to ABA, ethylene, auxin and BR[246]. These findings point to complex crosstalk mechanisms occurring between hormonal regulation and antioxidant responses in combinatorial stress acclimation. Taken together, when considering phytohormones as targets for improving combinatorial stress tolerance, complex interactions among different hormones must be considered. Perhaps for this reason, studies investigating the functions of specific hormones, such as ethylene, auxin, or gibberellic acid, in plant response to combinatorial stresses remain scarce and warrant further exploration.

## Future perspectives
It is clear that relative to the corresponding isolated stresses, co-occurring abiotic stresses usually cause distinct effects on plants and elicit unique acclimation responses (Fig. 1a, b). Acclimation strategies fitting a given combinatorial stress condition are determined by various factors. Unraveling the underlying complex mechanisms will require investigations in multiple dimensions connecting acclimation traits to tiers of gene regulation (Fig. 2) and must also involve interactions with the biotic (stress) environment. It must be noted that the terms 'resilience' and 'acclimation' used here, refer to morphological, developmental, molecular and physiological changes imposed by plants to cope with (combinatorial) abiotic stresses, rather than to traits that are of commercial significance such as seed yield or biomass production. When considering plant resilience in an agronomic context, it is first important to integrate yield traits with stress responsiveness across various levels. It is also important to consider how susceptibility to pathogens and interactions with beneficial microbes are affected. Beneficial microbes, such as rhizosphere bacteria, can mitigate stress effects by regulating the nutritional and hormonal balance in plants and inducing systemic tolerance to both biotic and abiotic stresses[247–249]. Integrating microbiome-based solutions with breeding and agronomic practices can be a sustainable approach to improve crop performance under increasingly variable environmental conditions. Such integration requires robust statistical and data analysis pipelines to accurately interpret the complex interactions between multiple stresses[10,20,153]. Moreover, to facilitate the transfer of knowledge from lab to field, closely mimicking stress severity (magnitude/dose) and combinations, as found in natural or agricultural settings in laboratory studies is essential. One approach to address this challenge is by using experimental setups that simulate field conditions. For example, González-García et al. developed the TGRooZ device. This allows shoots to experience heat stress while keeping root-zone temperatures closer to natural conditions, thus preventing excessive root heating that commonly occurs in laboratory experimental conditions[250]. In addition, the use of thermal gradient systems are useful to study stress combinations from a dose-response perspective[251]. Such advancements improve the physiological relevance of controlled experiments and improve the transferability of findings to real-world agricultural settings. Additionally, differential impact due to developmental (st)age, plant species, and stress cue hierarchy must be considered[105,252].

Plants exhibit considerable intra- and inter-species variation in response to abiotic stresses. Although such natural genetic and trait variation has been exploited to characterize acclimation mechanisms to diverse abiotic stresses[162,253–256], only few studies have done so for multi-stress resilience[6,235,257,258]. Investigating how natural genetic diversity translates into variation in combinatorial stress acclimation at the phenotypic level allows the identification of novel genes shaping typical traits that contribute to local adaptation, especially when the given stress combination is at a sublethal severity. Such investigations, especially when combined with integrative

~omics and/or functional genetics approaches[258,259], can yield invaluable knowledge towards the development of multi-stress resilience crops.

Future research should also focus on how combinatorial stresses affect productivity in different plant species, including agronomically-relevant crops, during more naturally relevant stress conditions, such as abiotic stresses at a sublethal severity[4,6,163]. Given that plant responses to environmental cues occur in a dose-dependent manner, acclimation strategies for abiotic stresses at milder severity might differ from those under more severe conditions. Finally, it is important to determine whether the multi-stress regulators and traits identified can contribute to the breeding and engineering of climate change-ready field crops. Following initial laboratory findings, extensive field testing is crucial to validate relevance of identified resilience traits in agro-environments. Identification of orthologues that share functionality and a common ancestor is often regarded as a first step for the knowledge transfer[260] from model species to crops. Coupled with genome editing techniques (CRISPR/Cas9), it will be hopefully possible to develop modern agricultural crops that possess broad resilience to multiple combinatorial stresses.

## Data availability
No new datasets were analyzed or generated in this review article.

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

## Acknowledgements
This research was funded by China Scholarship Council (CSC) grant 201806170025 to Z.J. and Nederlandse Organisatie voor Wetenschappelijk Onderzoek (NWO) grant OCENW.M20.197 to R.S. and 867.15.031 to R.S. and M.v.Z. We thank Rens Voesenek and Sjef Smeekens for their support in the early stages of the project.

## Author contributions
Z.J.: Writing -original draft; Writing -review & editing; R.S. M.v.Z.: Writing -review & editing.

## Competing interests
The authors declare no competing interests.
