## [Transparent Peer Review file · Communications Biology]

Mechanisms of plant acclimation to multiple abiotic stresses

Corresponding Author: Professor Rashmi Sasidharan

Version 0:

Reviewer comments:

Reviewer #1

(Remarks to the Author)

The subject of this review is very important. The authors first briefly discuss single stress conditions and then address the subject of stress combination. The authors provide a very comprehensive and balanced coverage of past work and overall, the manuscript reads well. In general, this is a very good review that pending English editing and a few revisions, could be accepted for publication.

Points to improve:

1. English. Please go through all the manuscript and perform English editing.
2. When discussing Differential Transpiration (line 179) please refrain from calling it 'deviant stomatal regulation'. Differential transpiration is a newly discovered acclimation strategy of plants to stress combination. Moreover, it is not a 'special case' it was also recently described in tomato (Bjerring Jensen, N., Vrobel, O., Akula Nageshbabu, N., De Diego, N., Tarkowski, P., Ottosen, C.O. and Zhou, R., 2024. Stomatal effects and ABA metabolism mediate differential regulation of leaf and flower cooling in tomato cultivars exposed to heat and drought stress. *Journal of Experimental Botany*, 75(7), pp.2156-2175.).
3. Also, regarding stomatal responses to drought and heat combination, please refer to this newly published work (Xu X, Liu H, Praat M, Pizzio GA, Jiang Z, Driever SM, Wang R, Van De Cotte B, Villers SLY, Gevaert K, Leonhardt N, Nelissen H, Kinoshita T, Vanneste S, Rodriguez PL, van Zanten M, Vu LD, De Smet I. Stomatal opening under high temperatures is controlled by the OST1-regulated TOT3-AHA1 module. *Nat Plants*. 2024 Nov 29. doi: 10.1038/s41477-024-01859-w. Epub ahead of print. PMID: 39613896.).
4. When discussing ROS and redox in plant stress responses, please cite this paper (Mittler R, Zandalinas SI, Fichman Y, Van Breusegem F. Reactive oxygen species signalling in plant stress responses. *Nat Rev Mol Cell Biol*. 2022 Oct;23(10):663-679. doi: 10.1038/s41580-022-00499-2. Epub 2022 Jun 27. PMID: 35760900.).
5. When describing the multifactorial stress combination concept, please also cite this work (Zandalinas SI, Mittler R. Plant responses to multifactorial stress combination. *New Phytol*. 2022 May;234(4):1161-1167. doi: 10.1111/nph.18087. Epub 2022 Mar 26. PMID: 35278228.), as well as this (Zandalinas SI, Fritschi FB, Mittler R. Global Warming, Climate Change, and Environmental Pollution: Recipe for a Multifactorial Stress Combination Disaster. *Trends Plant Sci*. 2021 Jun;26(6):588-599. doi: 10.1016/j.tplants.2021.02.011. Epub 2021 Mar 18. PMID: 33745784.).
6. In line 635 the authors highlight the need to study the effect of stress combination on plant productivity. Please cite (Peláez-Vico MÁ, Sinha R, Induri SP, Lyu Z, Venigalla SD, Vasireddy D, Singh P, Immadi MS, Pascual LS, Shostak B, Mendoza-Cózatl D, Joshi T, Fritschi FB, Zandalinas SI, Mittler R. The impact of multifactorial stress combination on reproductive tissues and grain yield of a crop plant. *Plant J*. 2024 Mar;117(6):1728-1745. doi: 10.1111/tpj.16570. Epub 2023 Dec 4. PMID: 38050346.).

Reviewer #2

(Remarks to the Author)

Combined stresses elicit unique responses involving complex mechanisms. This review highlights plant acclimation to co-

occurring abiotic stresses, emphasizing morphological, physiological, developmental, and molecular aspects. One main point in this review is discussion about key transcription factors (e.g., MBF1c, HSF, MYB, WRKY, NAC, ERF) regulating multi-stress responses, while miRNAs, heat shock proteins, and MAPKs (e.g., MAPK3/4/6) modulate post-transcriptional and translational processes. Further, epigenetic factors, hormonal regulation (e.g., ABA, auxin, JA, SA, BRs), and reactive oxygen species drive systemic stress signaling are also listed out.

This article is useful and I recommend the publication, but after a thorough major revision to fully address the comments below.

One conspicuous absence in this review, which cannot be left unnoticed, is the SCIPdb resource (www.nipgr.ac.in/scipdb.php, see PMID: 37824297), and this should be cited. Specific references include the first paragraph (lines 20-33), line 159 (several such traits are reported here), line 371 (transcriptomes), and others. Like the content mentioned in lines 30-33, several studies are listed in this database and comprehensively curated. This resource covers possible stress combinations and their practical implications. Similar to Figure 1, SCIPdb presents a large amount of phenomics data for all stress combinations reported so far. This contribution should be acknowledged. Overall, this resource, the only large-scale database available to date for combined stresses, warrants a few sentences in the manuscript to draw the attention of researchers.

In lines 360-366, it is worth noting that these analyses, in a preliminary way, can be performed using this database. The source data, hosted alongside the pipeline provided, are useful for researchers aiming to conduct specific analyses. Additionally, in lines 350-356, the availability of multi-omics data in one place, systematically organized, is a significant feature of this resource. Lines 270-282 also highlight a much more comprehensive stress matrix provided by SCIPdb, which is interactive and valuable for research.

In lines 14-16, for this information to be effectively reflected in the manuscript, it should be covered. Currently, it is unclear how this could be achieved. If the authors lack space to include this, the abstract should be modified accordingly.

The concept of acclimation in protecting plants from subsequent stresses is introduced in this review. Although this concept is well-known, it has not been discussed in the context of combined stresses. Highlighting this aspect could inspire useful research directions for future experiments. In fact, cross-protection due to acclimation with one stress followed by exposure to another stress is a promising idea worth exploring. The section starting at line 321 is useful but requires additional information beyond *Arabidopsis*.

Figure 2 does not appear to introduce pathways that differ significantly from single-stress pathways. However, given the limited understanding of the molecular mechanisms underlying plant responses to combined stresses, the figure is still relevant. The caption should clearly indicate whether the signaling depicted in the figure pertains solely to abiotic-abiotic stress combinations. The legend should also specify the stress combinations covered. Generalizing the pathway candidates for biotic stress combinations is inappropriate for this article, which does not aim to address biotic stresses comprehensively.

Figure 1 seems to be based entirely on *Arabidopsis* data. If so, the authors should clarify how these findings can be extrapolated to crop plants, particularly in relation to acclimation responses. The heat map is difficult to interpret—are these relative values? If so, what are the source values from actual experiments? Normalizing multiple types of parameters is challenging and may lead to misleading conclusions. This should be carefully examined.

Line 561 and the subsequent sentences offer overly obvious points that are unnecessary for a specialized review. These sections are also accompanied by excessive citations, adding to an already lengthy reference list.

In line 570, the discussion of hormones is minimal and lacks depth. This section offers a peripheral overview that does not sufficiently engage with the topic's complexities. The authors should provide a more focused synthesis of the content here.

Lines 46-51 are not well-covered in the manuscript. The figures also lack coherence. Including an exclusive figure on specific target combinations or expanding Figure 2 to provide deeper coverage of such combinations would enhance the manuscript.

The section beginning at line 53, which discusses single abiotic stresses, is largely unnecessary. There is already an extensive body of literature and reviews available on this topic. While it appears the authors aimed to connect single stresses to co-occurring stresses, the reliance on single-stress literature is not justified. The combined stress section that follows stands well on its own, rendering this introductory section redundant. Is Figure 1 also based on single-stress data?

While the authors correctly highlight ABA's key role, the details are not explicit. Specifically, the manuscript should discuss how ABA crosstalk with other hormones modulates signaling under combined stresses compared to single stresses. This interaction is increasingly supported by evidence and plays a crucial role in determining trade-offs. The authors should explore this aspect further.

Reviewer #3

(Remarks to the Author)

This review by Zhang Jiang, Martijn van Zanten, and Rashmi Sasidharan, titled "Mechanisms of Plant Acclimation to Multiple Abiotic Stresses," emphasizes the necessity of studying combinations of multiple stresses. In nature, abiotic, and

also biotic, stresses rarely occur in isolation; they typically appear in combination. This necessity has become urgent due to the effects of climate change, as the negative impacts of abiotic stresses are becoming more frequent and severe. Consequently, we are entering a new scenario in plant adaptation to the combined effects of multiple stresses. The review highlights an important message: the effects of combined stresses are not merely the sum of individual ones. In some combinations, the effect of a second stress can be additive or might counteract the negative impact of the first one. I enjoyed reading this review and, overall, I consider it very interesting for the plant community. This review is timely and appropriate given the current climate change scenario and the need for developing more sustainable agriculture. Therefore, it is crucial to understand how plants respond to combinations of multiple stresses at different levels. I find the organization of the review to be appropriate, as it analyzes responses at various levels (transcriptomic, translational, hormonal, etc.).

Nonetheless, I have a few comments:

- The title states: "Acclimation to multiple abiotic stresses." However, the authors primarily mention heat and drought when discussing stress combinations. They should review other stress combinations that have been published. A recent review by Sanchez et al. (DOI: 10.3389/fpls.2022.918537) discusses several stress combinations, although focusing more on root effects. This article should be referenced.
- They should comment on the low correlation between transcriptional and translational responses in response to stress (or combinations)
- They highlight that the negative effect of stress combinations is more dramatic during the reproductive stage than during vegetative growth. However, I do not completely agree with this statement. Although the reproductive stage is essential in agricultural production, the negative effects of stresses during seedling establishment and root system formation are crucial for plant survival and productivity. This statement should be reconsidered.
- In the "Future Perspectives" section, they include the sentence: "Moreover, to facilitate the transfer of knowledge from lab to field, closely mimicking stress severity (magnitude/dose) and combinations as found in natural or agricultural settings in laboratory studies is essential." As they emphasized the effects of heat stress, I was surprised that they did not include a comment on this paper (DOI: 10.1016/j.xplc.2022.100514), where the authors showed an interesting device to analyze heat stress mimicking natural conditions in the laboratory to avoid excessive heat in the root system.
- I miss some comments on the use of microbiomes as a tool to improve multi-stress resilience in crops.

Version 1:

Reviewer comments:

Reviewer #1

(Remarks to the Author)

The authors have answered all of my comments.

Reviewer #2

(Remarks to the Author)

In the revised manuscript, the authors have addressed most of my comments. I have also reviewed the other two reviewers' comments, and it seems the authors have addressed them as well. Among my comments, two aspects that could benefit from further clarification are:

The synthesis and deeper analysis of hormone roles.

The mixing of single-stress information to connect with combined stresses, as such information cannot be directly extrapolated to unique combined-stress responses.

For these two points, the authors could provide notes in the appropriate sections and proceed.

I recommend this manuscript for publication.

Reviewer #3

(Remarks to the Author)

The review entitled "Mechanisms of Plant Acclimation to Multiple Abiotic Stresses" by Jiang et al. addresses a critical topic in plant biology, as stress combinations naturally occur in plant ecosystems and significantly impact crop productivity. The authors deliver a thorough and well-balanced analysis of the effects of both individual and combined stresses on plant growth and development across multiple levels. They have demonstrated considerable effort in revising the manuscript, thoughtfully addressing the majority of the reviewers' comments and suggestions. Notably, as recommended by Reviewer 2, they incorporated the SCIPdb resource—a valuable tool for analyzing and understanding the impacts of stress combinations on plants. Overall, this review is highly relevant and insightful for the plant research community, and I strongly recommend it for publication in *Communications Biology*.

Response to reviewers

(LINE Numbers mentioned below correspond to those in the 'Marked file' version)

Reviewer #1 (Remarks to the Author):

The subject of this review is very important. The authors first briefly discuss single stress conditions and then address the subject of stress combination. The authors provide a very comprehensive and balanced coverage of past work and overall, the manuscripts reads well. In general, this is a very good review that pending English editing and a few revisions, could be accepted for publication.

Points to improve:

1) English. Please go through all the manuscript and perform English editing.

>> **Response:** We have carefully revised the manuscript to improve the clarity, grammar, and overall readability. The text has been edited to improve sentence structure, conciseness and flow, while maintaining the scientific accuracy. We have also shortened text in several places.

2) When discussing Differential Transpiration (line 179) please refrain from calling it 'deviant stomatal regulation'. Differential transpiration is a newly discovered acclimation strategy of plants to stress combination. Moreover, it is not a 'special case' it was also recently described in tomato (Bjerring Jensen, N., Vrobel, O., Akula Nageshbabu, N., De Diego, N., Tarkowski, P., Ottosen, C.O. and Zhou, R., 2024. Stomatal effects and ABA metabolism mediate differential regulation of leaf and flower cooling in tomato cultivars exposed to heat and drought stress. *Journal of Experimental Botany*, 75(7), pp.2156-2175.).

>> **Response:** We have modified the terminology from 'deviant stomatal regulation' to 'differential transpiration' (Line 185). We have now cited Bjerring Jensen *et al.* (2024) (Reference 112) in the section discussing stomatal responses (Line 184).

3) Also, regarding stomatal responses to drought and heat combination, please refer to this newly published work (Xu X, Liu H, Praat M, Pizzio GA, Jiang Z, Driever SM, Wang R, Van De Cotte B, Villers SLY, Gevaert K, Leonhardt N, Nelissen H, Kinoshita T, Vanneste S, Rodriguez PL, van Zanten M, Vu LD, De Smet I. Stomatal opening under high temperatures is controlled by the OST1-regulated TOT3-AHA1 module. *Nat Plants*. 2024 Nov 29. doi: 10.1038/s41477-024-01859-w. Epub ahead of print. PMID: 39613896.).

>> **Response:** This reference to our work was already cited in the paper as a pre-print version (De Smet et al). As the paper is now published in *Nature Plants*, we have included the

reference to the final published version (of Xu *et al.* 2024) (Reference 107) to replace the referral to the preprint version.

4) When discussing ROS and redox in plant stress responses, please cite this paper (Mittler R, Zandalinas SI, Fichman Y, Van Breusegem F. Reactive oxygen species signalling in plant stress responses. *Nat Rev Mol Cell Biol.* 2022 Oct;23(10):663-679. doi: 10.1038/s41580-022-00499-2. Epub 2022 Jun 27. PMID: 35760900.).

>> **Response:** We have incorporated the work by Mittler *et al.* (2022) (Reference 129) in the section on 'ROS production and signaling' as requested.

5) When describing the multifactorial stress combination concept, please also cite this work (Zandalinas SI, Mittler R. Plant responses to multifactorial stress combination. *New Phytol.* 2022 May;234(4):1161-1167. doi: 10.1111/nph.18087. Epub 2022 Mar 26. PMID: 35278228.), as well as this (Zandalinas SI, Fritschi FB, Mittler R. Global Warming, Climate Change, and Environmental Pollution: Recipe for a Multifactorial Stress Combination Disaster. *Trends Plant Sci.* 2021 Jun;26(6):588-599. doi:10.1016/j.tplants.2021.02.011. Epub 2021 Mar 18. PMID: 33745784.).

>> **Response:** We do not understand this remark as Zandalinas *et al.* (2022) (Reference 11) and Zandalinas *et al.* (2021) (Reference 166) were already cited in relevant sections (e.g. lines 33, 184, 282, 308, 310).

6) In line 635 the authors highlight the need to study the effect of stress combination on plant productivity. Please cite (Peláez-Vico MÁ, Sinha R, Induri SP, Lyu Z, Venigalla SD, Vasireddy D, Singh P, Immadi MS, Pascual LS, Shostak B, Mendoza-Cózatl D, Joshi T, Fritschi FB, Zandalinas SI, Mittler R. The impact of multifactorial stress combination on reproductive tissues and grain yield of a crop plant. *Plant J.* 2024 Mar;117(6):1728-1745. doi: 10.1111/tbj.16570. Epub 2023 Dec 4. PMID: 38050346.).

>> **Response:** We have now cited Peláez-Vico *et al.* (2024) (Reference 163) in the relevant section as requested.

Reviewer #2 (Remarks to the Author):

Combined stresses elicit unique responses involving complex mechanisms. This review highlights plant acclimation to co-occurring abiotic stresses, emphasizing morphological, physiological, developmental, and molecular aspects. One main point in this review is

discussion about key transcription factors (e.g., MBF1c, HSF, MYB, WRKY, NAC, ERF) regulating multi-stress responses, while miRNAs, heat shock proteins, and MAPKs (e.g., MAPK3/4/6) modulate post-transcriptional and translational processes. Further, epigenetic factors, hormonal regulation (e.g., ABA, auxin, JA, SA, BRs), and reactive oxygen species drive systemic stress signaling are also listed out. This article is useful and I recommend the publication, but after a thorough major revision to fully address the comments below.

1) One conspicuous absence in this review, which cannot be left unnoticed, is the SCIPdb resource (www.nipgr.ac.in/scipdb.php, see PMID: 37824297), and this should be cited. Specific references include the first paragraph (lines 20-33), line 159 (several such traits are reported here), line 371 (transcriptomes), and others. Like the content mentioned in lines 30-33, several studies are listed in this database and comprehensively curated. This resource covers possible stress combinations and their practical implications. Similar to Figure 1, SCIPdb presents a large amount of phenomics data for all stress combinations reported so far. This contribution should be acknowledged. Overall, this resource, the only large-scale database available to date for combined stresses, warrants a few sentences in the manuscript to draw the attention of researchers.

In lines 360-366, it is worth noting that these analyses, in a preliminary way, can be performed using this database. The source data, hosted alongside the pipeline provided, are useful for researchers aiming to conduct specific analyses. Additionally, in lines 350-356, the availability of multi-omics data in one place, systematically organized, is a significant feature of this resource. Lines 270-282 also highlight a much more comprehensive stress matrix provided by SCIPdb, which is interactive and valuable for research.

>> **Response:** We fully agree with the reviewer of the importance of acknowledging this very important resource. We have incorporated citations to the SCIPDb database (Reference 13) in multiple sections, including Lines 13, 33,165,372,383,385,401 as requested. Additionally, we replaced the GO and KEGG discussion with a brief description of SCIPDb (Lines 380-396).

2) In lines 14-16, for this information to be effectively reflected in the manuscript, it should be covered. Currently, it is unclear how this could be achieved. If the authors lack space to include this, the abstract should be modified accordingly.

>>**Response:** Discussing knowledge transfer to field is not the main scope of our review. So, we agree with the reviewer and have generalized the last sentence of the abstract.

3) The concept of acclimation in protecting plants from subsequent stresses is introduced in this review. Although this concept is well-known, it has not been discussed in the context of combined stresses. Highlighting this aspect could inspire useful research directions for future

experiments. In fact, cross-protection due to acclimation with one stress followed by exposure to another stress is a promising idea worth exploring. The section starting at line 321 is useful but requires additional information beyond Arabidopsis.

>>**Response:** We agree that cross- acclimation is important and should be emphasized in our manuscript. In Line 324, we already included the discussion of cross-acclimation in poplar trees. We have now modified the section (Line 318 onwards). Additionally, application of cross-acclimation in crops is also highlighted within the "Epigenetic Regulation" section (Lines 549 onwards)

4) Figure 2 does not appear to introduce pathways that differ significantly from single-stress pathways. However, given the limited understanding of the molecular mechanisms underlying plant responses to combined stresses, the figure is still relevant. The caption should clearly indicate whether the signaling depicted in the figure pertains solely to abiotic-abiotic stress combinations. The legend should also specify the stress combinations covered. Generalizing the pathway candidates for biotic stress combinations is inappropriate for this article, which does not aim to address biotic stresses comprehensively.

>> **Response:** We agree with the reviewer's comment that the involvement of the presented single-stress signalling and perception pathways is to a large extent speculative. As suggested, we have now emphasized this in the caption and legend text.

As suggested, we removed the referral to abiotic-biotic stress interactions by removing the phrase on the role of *miRNA16C* in resilience to combined drought and bacterial infection.

Also as requested, in the legend we now mention the (putative) combinatorial stress pathway wherein a specific mentioned pathways are involved in, based on the main text description.

5) Figure 1 seems to be based entirely on Arabidopsis data. If so, the authors should clarify how these findings can be extrapolated to crop plants, particularly in relation to acclimation responses. The heat map is difficult to interpret—are these relative values? If so, what are the source values from actual experiments? Normalizing multiple types of parameters is challenging and may lead to misleading conclusions. This should be carefully examined.

>> **Response:** Indeed, most of the physiological and molecular traits in Panel A of Figure 1 are derived from Arabidopsis data, except for the stomatal regulation in flowers and leaves, which is from a soybean study. We think this is sufficiently clear from the legend (and the plant picture in the figure), but added an explicit mention to Arabidopsis nonetheless in the Panel A description. As requested, we clarified that the findings can be transferred to crop studies by adjusting the text in the figure legend.

The presented heat map values in panel B are indeed relative. We revised the legend of Panel B allowing better interpretation of the heat map and added the relevant references, as requested.

6) Line 561 and the subsequent sentences offer overly obvious points that are unnecessary for a specialized review. These sections are also accompanied by excessive citations, adding to an already lengthy reference list.

>> **Response:** We agree with the comment and have condensed the paragraph and references about flavonoids (Line 603 onwards).

7) In line 570, the discussion of hormones is minimal and lacks depth. This section offers a peripheral overview that does not sufficiently engage with the topic's complexities. The authors should provide a more focused synthesis of the content here.

>> **Response:** We do not agree that this section is too peripheral. As mentioned in this review, plant hormones have received a lot of attention in recent years in the context of multi-stress resilience. Especially ABA is considered important in the responses to stress combinations such as salt combined with heat (study on Arabidopsis by Suzuki *et al.*, 2016, Reference 159). Accordingly, we put ABA in the spotlight for an in-depth discussion in this section on plant hormones, but also incorporated other hormones like JA and SA to compare their roles in multi-stress responses.

8) Lines 46-51 are not well-covered in the manuscript. The figures also lack coherence. Including an exclusive figure on specific target combinations or expanding Figure 2 to provide deeper coverage of such combinations would enhance the manuscript.

>> **Response:** We initially attempted to generate such an integrated figure with expanded information. However, we realized that it is not feasible to have one figure legibly depict this information or lists of responses to all types of stress combinations. Such a figure became quickly incomprehensible, also because interpretation needs context (species, dose severity, type of stress application, indoor or field study etc.), which is not easily captured in one figure. Therefore, we chose to focus on Arabidopsis as a case example. However we welcome more detailed suggestions and inputs from the editor or reviewers on how to improve the figure to be more comprehensive, without becoming overwhelming or out of context.

9) The section beginning at line 53, which discusses single abiotic stresses, is largely unnecessary. There is already an extensive body of literature and reviews available on this topic. While it appears the authors aimed to connect single stresses to co-occurring stresses,

the reliance on single-stress literature is not justified. The combined stress section that follows stands well on its own, rendering this introductory section redundant. Is Figure 1 also based on single-stress data?

>>**Response:** Figure 1 is based on data of both single-stress and combined-stress studies, as also indicated in the header and panels and legend text. We indeed chose to start with summarizing knowledge on single stress acclimation mechanisms. We deem this necessary to provide the minimum background of single stresses essential to understand multi-stress concepts, but also to bring the point across that (and how) single stress acclimation differs from less studied acclimation responses to multi-stress. It is our conviction that discussion of single-stress responses first, by introducing the main signalling and response concepts, facilitates a better understanding of the later-on introduced combined-stress acclimation responses (that are understudied).

10) While the authors correctly highlight ABA's key role, the details are not explicit. Specifically, the manuscript should discuss how ABA crosstalk with other hormones modulates signaling under combined stresses compared to single stresses. This interaction is increasingly supported by evidence and plays a crucial role in determining trade-offs. The authors should explore this aspect further.

>>**Response:** As suggested, we have added some more details on ABA crosstalk with other hormones and now better discuss the trade-off regulation of ABA under combined heat and drought stress (Lines 640 onwards).

Reviewer #3 (Remarks to the Author):

This review by Zhang Jiang, Martijn van Zanten, and Rashmi Sasidharan, titled "Mechanisms of Plant Acclimation to Multiple Abiotic Stresses," emphasizes the necessity of studying combinations of multiple stresses. In nature, abiotic, and also biotic, stresses rarely occur in isolation; they typically appear in combination. This necessity has become urgent due to the effects of climate change, as the negative impacts of abiotic stresses are becoming more frequent and severe. Consequently, we are entering a new scenario in plant adaptation to the combined effects of multiple stresses. The review highlights an important message: the effects of combined stresses are not merely the sum of individual ones. In some combinations, the effect of a second stress can be additive or might counteract the negative impact of the first one.

I enjoyed reading this review and, overall, I consider it very interesting for the plant community. This review is timely and appropriate given the current climate change scenario

and the need for developing more sustainable agriculture. Therefore, it is crucial to understand how plants respond to combinations of multiple stresses at different levels. I find the organization of the review to be appropriate, as it analyzes responses at various levels (transcriptomic, translational, hormonal, etc.).

Nonetheless, I have a few comments:

1) The title states: "Acclimation to multiple abiotic stresses." However, the authors primarily mention heat and drought when discussing stress combinations. They should review other stress combinations that have been published. A recent review by Sanchez et al. (DOI: 10.3389/fpls.2022.918537) discusses several stress combinations, although focusing more on root effects. This article should be referenced.

>>**Response:** While heat and drought are indeed the most discussed stress combination in our work, we have also discussed other stress interactions, including heat and salt (References 159 and 219), heat and high light (References 113 and 230) and flooding followed by drought (References 4 and 6).

As suggested we now have included a citation to Sanchez *et al.* (2022) to incorporate additional perspectives on root-mediated responses to multiple stress conditions (Reference 106; Line 168, 228 and 396).

2) They should comment on the low correlation between transcriptional and translational responses in response to stress (or combinations)

>>**Response:** We agree with the reviewer's comment and have added the following text at the start of the 'Post-transcriptional regulation' section (Line 463).

3) They highlight that the negative effect of stress combinations is more dramatic during the reproductive stage than during vegetative growth. However, I do not completely agree with this statement. Although the reproductive stage is essential in agricultural production, the negative effects of stresses during seedling establishment and root system formation are crucial for plant survival and productivity. This statement should be reconsidered.

>>**Response:** We believe the reviewer refers to the text in Line 195-197 where we stated:

'For instance, the negative impact of combined heat and drought on plant yield is more pronounced if the stress combination happens during the reproductive stage than when it occurs during vegetative growth.'

We agree with the reviewer and have removed the statement.

4) In the "Future Perspectives" section, they include the sentence: "Moreover, to facilitate the transfer of knowledge from lab to field, closely mimicking stress severity (magnitude/dose) and combinations as found in natural or agricultural settings in laboratory studies is essential." As they emphasized the effects of heat stress, I was surprised that they did not include a comment on this paper (DOI: 10.1016/j.xplc.2022.100514), where the authors showed an interesting device to analyze heat stress mimicking natural conditions in the laboratory to avoid excessive heat in the root system.

>>**Response:** We have included the proposed paper; González-García et al. (2023; Reference 250), emphasizing its relevance in improving the physiological relevance of controlled experiments, and also included a mention on thermal gradient setups (Reference 251) (Line 694 onwards)

5) I miss some comments on the use of microbiomes as a tool to improve multi-stress resilience in crops.

>>**Response:** We have added a short discussion on the role of microbiomes in improving plant resilience to stress combinations in Lines 683-690.

Response to reviewer comments:

We thank the reviewers for their kind remarks and suggestions for improving this manuscript.

Reviewer 1 and 3 had no further remarks.

Reviewer 2 requested clarification for two aspects:

1) “Synthesis and deeper analysis of hormone roles”

The following sentences in the text (“Hormonal interactions”) clarifies this: *Taken together, when considering phytohormones as targets for improving combinatorial stress tolerance, complex interactions among different hormones must be considered. Perhaps for this reason, studies investigating the functions of specific hormones, such as ethylene, auxin, or gibberellic acid, in plant response to combinatorial stresses remain scarce and warrant further exploration.*

2) “The mixing of single-stress information to connect with combined stresses, as such information cannot be directly extrapolated to unique combined-stress responses”.

This has been stated in the abstract in lines 152-156 in the section “Effects of combined abiotic stresses on plant functioning” followed by several comprehensive examples and repetition in other sections. We therefore feel it would not be useful to restate this concept yet again (also considering we are at the word limit).